# DPGNet: Modeling Dynamic Graphs and Complex Temporal Patterns for Spatiotemporal Forecasting

## Abstract

Spatiotemporal prediction plays a crucial role in various domains, such as traffic management and weather forecasting. Nevertheless, existing spatiotemporal prediction models remain limited in their ability to capture dynamic inter-node relationships and to comprehensively model complex temporal patterns. To address these limitations, we propose the Dynamic Graph Prediction Network (DPGNet), which includes two key components: the Adaptive Graph Learner (AGL) and the Adaptive Season Learner (ASL). AGL is a plug-and-play module that can effectively capture dynamic inter-node relationships while suppressing weak connections. ASL can model the complex temporal patterns of nodes and construct graph structures across different temporal patterns and time scales. We conduct extensive experiments on five real-world spatiotemporal forecasting datasets, and the results demonstrate that DPGNet outperforms many state-of-the-art methods in both effectiveness and efficiency. Our code is accessible at https://anonymous.4open.science/r/DPGNet.

## 1 Introduction

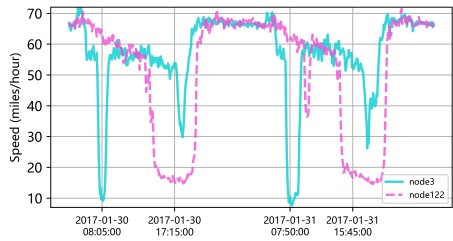 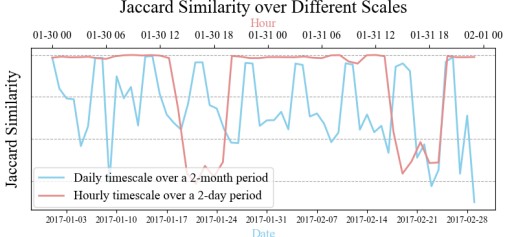

(a) The traffic flow of 2 different nodes.     (b) Correlations under different scales

Figure 1: The traffic flow and node correlations in the PEMS-Bay dataset. (a) shows the traffic flow of two nodes, which clearly display distinct periodic patterns. (b) presents the inter-node correlations computed using sliding-window Jaccard similarity with different time scales.

Spatiotemporal forecasting has attracted widespread attention across various fields, including traffic management (Guo et al., 2023), climate prediction modeling (Jones, 2017), and epidemic spread modeling (Luo et al., 2025). Although existing spatiotemporal forecasting models have achieved remarkable success (Liu & Zhang, 2024; Qian & Mallick, 2024; Lee & Ko, 2024), they still struggle to fully capture the complex temporal patterns inherent in spatiotemporal data. As shown in Figure 1 (a), each node exhibits complex temporal patterns (Bai et al., 2020; Guo et al., 2021), primarily driven by the varied and intertwined trend and seasonal components (Wu et al., 2021) and the multi-scale structures (Wang et al., 2024). The tight coupling of complex temporal patterns with spatial structure poses an additional challenge. As illustrated in Figure 1 (b), the correlations between nodes not only vary over time but also differ across time scales. Existing models struggle to capture dynamic correlations between nodes (Deng et al., 2021; Gao et al., 2024), and often learn numerous weak connections (Wu et al., 2019; Gao et al., 2024). For example, the value distribution of the adjacency

matrix generated by (Wu et al., 2019) is shown in Figure 2, which reveals numerous weak connections (i.e., low-value entries). ST-Norm (Deng et al., 2021) posits that these weak connections are caused by noise and provides limited benefit to predictive accuracy.

To address these challenges, we propose the Dynamic Graph Prediction Network (DPGNet), which comprises two core components: Adaptive Season Learner (ASL) and Adaptive Graph Learner (AGL). ASL combines a temporal decomposition strategy (Wu et al., 2021), a multiscale processing approach (Wang et al., 2024), and a pattern- and scale-specific graph construction method. This component strengthens the model's ability to capture complex temporal patterns while decoupling spatial information from those temporal patterns across different time scales. AGL integrates self-attention with gating mechanisms to effectively capture dynamic inter-node relationships and suppress weak connections. In summary, our contributions are as follows:

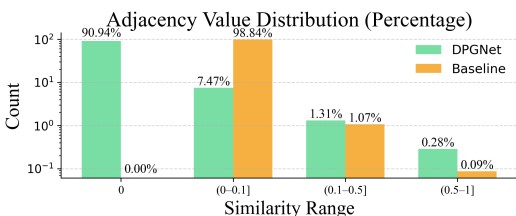

Figure 2: The learned adjacency matrix of the model on METR-LA. Adjacency values within the range of $(0, 0.1]$ are considered weak connections.

- We propose a novel graph structure generator, AGL, which effectively captures dynamic inter-node relationships while suppressing weak connections. Its plug-and-play design allows it to be seamlessly integrated into existing models to enhance prediction accuracy.
- We introduce a spatiotemporal pattern learning module, ASL, which models complex temporal dynamics of nodes and constructs graph structures across different temporal patterns and time scales.
- Extensive experiments demonstrate that DPGNet consistently outperforms state-of-the-art baselines in both prediction accuracy and model efficiency.

The remainder of this paper is organized as follows. Section 2 reviews related work. Section 3 introduces the theoretical foundations and provides an overview of the proposed model. Section 4 presents experimental evaluations of the model. Finally, Section 5 concludes this article.

## 2 RELATED WORK

### 2.1 SPATIOTEMPORAL PREDICTION

Recent advancements in deep learning have significantly improved spatiotemporal prediction. For example, DCRNN (Li et al., 2017) enhances Recurrent Neural Networks (RNNs) by using diffusion convolution to capture spatiotemporal patterns. Building on this, AGCRN (Bai et al., 2020) introduces Node Adaptive Parameter Learning (NAPL) to boost model adaptability. To address RNN issues like gradient vanishing (Noh, 2021) and error accumulation (Jenckel et al., 2017), STGCN (Yan et al., 2018) employs Temporal Convolutional Networks (TCNs) instead. GWNet (Wu et al., 2019) uses an encoder-decoder framework, combining TCNs and GCNs in the encoder to model temporal and spatial dependencies, while the decoder maps historical data to future sequences using Fully Connected (FC) layers. However, TCNs struggle with long-term patterns due to limited receptive fields (Fan et al., 2023; Luo & Wang, 2024). Transformers (Vaswani et al., 2017) overcome this with their attention mechanism, enabling global context modeling. For instance, DG2RNN (Zhao et al., 2022) integrates temporal and spatial attention into RNNs to learn both time-based and spatial relationships. Hybrid models have also shown strong performance. PMC-GCN (Gao et al., 2024) introduces a personalized enhanced GCN (P-GCN) with multi-head attention for better predictions. STIDGCN (Liu & Zhang, 2024) uses a pattern bank to store unique traffic patterns per node and incorporates spatiotemporal interactive learning. TESTAM (Lee & Ko, 2024) leverages the Mixture-of-Experts (MoE) framework for real-time traffic forecasting.

### 2.2 DYNAMIC IMPLICIT RELATIONSHIPS BETWEEN NODES

Implicit relationships, unlike physical connections, reflect correlations in node time series and are often dynamic and noisy. GWNet (Wu et al., 2019) captures these using learnable parameters. ST-Norm (Deng et al., 2021) further improves this with spatiotemporal embeddings and normalization to

reduce noise. To handle dynamic implicit relationships, MegaCRN (Jiang et al., 2023) introduces a Meta-Graph Learner, using memory cells to represent and dynamically update inter-node relationships based on input data. Similarly, PGCN (Shin & Yoon, 2024) dynamically adjusts graph structures during training to capture complex, time-varying spatial correlations.

### 2.3 TIME SERIES PROCESSING METHODS IN SPATIOTEMPORAL FORECASTING

Spatiotemporal data, essentially multivariate time series with spatial dependencies, can leverage various time series processing techniques for forecasting. These include temporal decomposition (Wu et al., 2021), multi-scale processing (Wang et al., 2024), Fourier transform method (Xu et al., 2024), and wavelet transform approaches (Sun et al., 2024; Qian & Mallick, 2024). Autoformer (Wu et al., 2021) introduces temporal decomposition to deep learning-based time series forecasting. It uses a moving average filter to split input sequences into trend and seasonal components, simplifying predictions. TimeMixer (Wang et al., 2024) creates a multi-scale framework by downsampling sequences to capture patterns at different scales, with a feature fusion mechanism to integrate these scales for comprehensive pattern recognition.

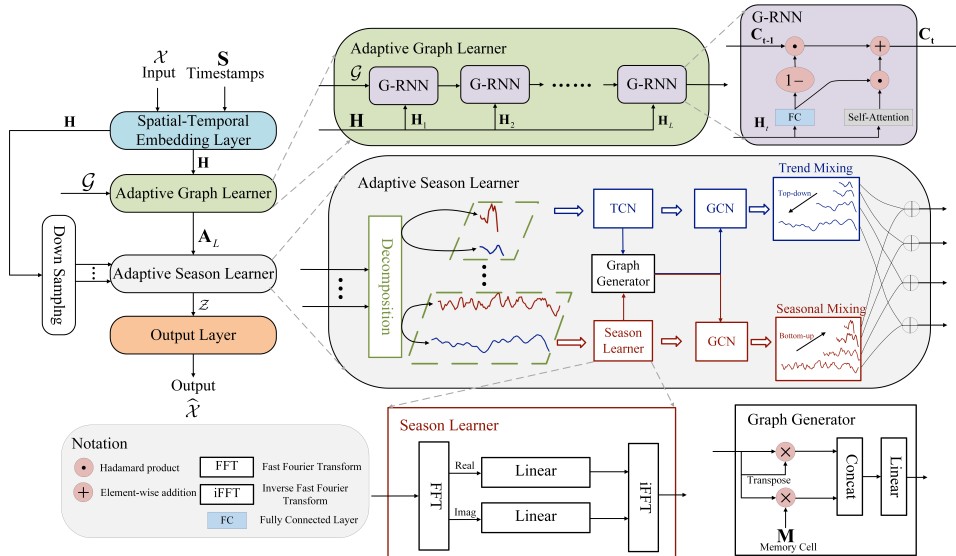

Figure 3: The overall architecture of DPGNet. Red lines and boxes indicate the processing of seasonal features, while blue lines and boxes represent the processing of trend features.

## 3 METHODOLOGY

To effectively capture dynamic implicit relationships among nodes and complex temporal patterns in spatiotemporal data, we propose DPGNet. In the following, we will provide an overview of the model architecture, followed by a detailed description of each core module.

### 3.1 OVERVIEW

Given a spatiotemporal dataset $\boldsymbol{\mathcal{X}} = (\boldsymbol{X}_1, \boldsymbol{X}_2, \ldots, \boldsymbol{X}_L) \in \mathbb{R}^{N \times L \times C}$, where $N$ is the number of nodes, $L$ is the sequence length, and $C$ is the feature dimension, the objective of DPGNet is to predict the future sequence $(\boldsymbol{X}_{L+1}, \boldsymbol{X}_{L+2}, \ldots, \boldsymbol{X}_{L+O}) \in \mathbb{R}^{N \times O \times C}$, based on $\boldsymbol{\mathcal{X}}$ and a directed graph $\mathcal{G} = (\mathcal{V}, \mathcal{E}, \boldsymbol{A})$, where $\mathcal{V}$ and $\mathcal{E}$ represent the sets of nodes and edges, respectively, and $\boldsymbol{A} \in \mathbb{R}^{N \times N}$ is the adjacency matrix encoding explicit node relationships. As shown in Figure 3, DPGNet consists of a Spatial-Temporal Embedding Layer, an AGL, an ASL, and an Output Layer. The Spatial-Temporal Embedding Layer encodes both timestamp and spatiotemporal data. The AGL is composed of $L$ stacked G-RNN blocks. Each G-RNN integrates a self-attention mechanism and a gating mechanism. The update gate incorporates features from current spatiotemporal data to update

implicit relationships, while the reset gate discards noise and weak connections in the graph. The ASL incorporates temporal decomposition and multi-scale processing. Temporal decomposition enables the model to focus on more detailed temporal patterns, such as seasonal and trend patterns. Multi-scale processing enables the model to focus on the macro-level trends of nodes, which benefits long-term forecasting. In addition, the ASL consists of a TCN and a Season Learner, which help capture spatial relationships under specific temporal patterns. The Output Layer is composed of linear layers, used to perform the prediction task. Next, we will detail the structure of each module.

## 3.2 SPATIAL-TEMPORAL EMBEDDING LAYER

To reduce non-stationarity in time series and differences in feature distribution across nodes, we first normalize the input data along both temporal and spatial dimensions and then embed it using a linear layer. Additionally, we also incorporate timestamps into the model. Specifically, given the input data $\boldsymbol{\mathcal{X}} \in \mathbb{R}^{N \times L \times C}$ and the corresponding timestamps $\boldsymbol{S} \in \mathbb{R}^{L \times M}$, where $M$ denotes the dimension of the timestamp features, the Spatial-Temporal Embedding Layer can be described as:

$$\boldsymbol{H} = Linear(Norm_T(\boldsymbol{\mathcal{X}})) + Linear(Norm_S(\boldsymbol{\mathcal{X}})) + Linear(\boldsymbol{S}), \tag{1}$$

where $Norm_T(\boldsymbol{\mathcal{X}})$ and $Norm_S(\boldsymbol{\mathcal{X}})$ denote normalizations applied to the temporal and spatial dimensions of the input data $\boldsymbol{\mathcal{X}}$, respectively. The resulting embedded spatiotemporal features are denoted as $\boldsymbol{H} = [\boldsymbol{H}_1, \cdots, \boldsymbol{H}_L] \in \mathbb{R}^{N \times L \times h}$, where $h$ is the predefined hidden dimension.

## 3.3 ADAPTIVE GRAPH LEARNER

To effectively capture dynamic, time-varying implicit relationships between nodes and mitigate the impact of noise-induced weak connections on the learning process, we propose the AGL. As shown in Figure 3, the AGL consists of $L$ G-RNN units. Each G-RNN integrates a gating mechanism with self-attention, and its hidden states dynamically track the evolution of the adjacency matrix throughout the graph construction process. During the G-RNN training process, the module incorporates implicit relationships extracted from the data into the adjacency matrix through the update gate, while discarding noise and weak connections in the graph through the reset gate. For the spatiotemporal features at time step $t$, $\boldsymbol{H}_t \in \mathbb{R}^{N \times h}$, the processing flow of the AGL is as follows:

$$\boldsymbol{Q} = \boldsymbol{H}_t \boldsymbol{W}_q + \boldsymbol{b}_q, \quad \boldsymbol{K} = \boldsymbol{H}_t \boldsymbol{W}_k + \boldsymbol{b}_k, \quad \boldsymbol{A}_t = \tanh(LN(\boldsymbol{Q}\boldsymbol{K}^\top / \sqrt{h})), \tag{2}$$

$$\boldsymbol{i} = \sigma(\boldsymbol{H}_t \boldsymbol{w}_i + \boldsymbol{b}_i), \quad \boldsymbol{f} = \boldsymbol{1} - \boldsymbol{i}, \quad \boldsymbol{C}_t = \sigma(\boldsymbol{f} \odot \boldsymbol{C}_{t-1} + \boldsymbol{i} \odot \boldsymbol{A}_t), \tag{3}$$

where $\boldsymbol{W}_q \in \mathbb{R}^{h \times h}$, $\boldsymbol{W}_k \in \mathbb{R}^{h \times h}$, $\boldsymbol{w}_i \in \mathbb{R}^h$, $\boldsymbol{b}_q \in \mathbb{R}^h$, $\boldsymbol{b}_k \in \mathbb{R}^h$, and $\boldsymbol{b}_i \in \mathbb{R}^N$ are all trainable parameters shared across all G-RNN units. $\tanh(\cdot)$ denotes the tanh activation function, $\sigma(\cdot)$ denotes the sigmoid activation function. $\boldsymbol{i}, \boldsymbol{f} \in \mathbb{R}^N$ represent the update gate and the reset gate, respectively. $\boldsymbol{1} \in \mathbb{R}^N$ is a vector of all ones. $\boldsymbol{Q}, \boldsymbol{K} \in \mathbb{R}^{N \times h}$ are the query and key matrices in the self-attention mechanism, and $LN(\cdot)$ denotes layer normalization. $\boldsymbol{A}_t \in \mathbb{R}^{N \times N}$ represents the implicit relationships between nodes at time step $t$, and $\boldsymbol{C}_t \in \mathbb{R}^{N \times N}$ is the hidden state at $t$, which integrates adjacency matrix information at each time step, thereby enhancing the model's interpretability. It is worth noting that $\boldsymbol{C}_0 = \boldsymbol{A}$, which represents explicit inter-node relationships. Based on this design, the update gate introduces dynamic implicit relationships into the adjacency matrix, while the reset gate removes the weak connections, helping the model suppress noise. The output $\boldsymbol{A}_L$ of the final G-RNN is used to guide the aggregation of spatiotemporal information in subsequent modules. To mitigate the error accumulation (Jenckel et al., 2017) associated with RNN, we apply patch processing (Nie et al., 2023) to the input $\boldsymbol{H} \in \mathbb{R}^{N \times L \times h}$. This approach partitions the sequence $\boldsymbol{H}$ of length $L$ into $b$ non-overlapping patches, each of fixed length $p$, yielding $\boldsymbol{H_p} \in \mathbb{R}^{N \times b \times (p \times h)}$. We then apply a linear projection to map the feature dimension $(p \times h)$ of $\boldsymbol{H_p}$ to $h$. This approach reduces the number of G-RNN from $L$ to $b$ ($b \ll L$), effectively alleviating the error accumulation.

## 3.4 ADAPTIVE SEASON LEARNER

To improve the model's capability in extracting complex temporal patterns of nodes, we introduce multi-scale processing and temporal decomposition. Multi-scale processing captures characteristics of the spatiotemporal sequence at different scales (Mozer, 1991). Temporal decomposition helps decompose complex time series patterns (Wu et al., 2021), which simplifies the modeling process.

We first construct a multi-scale view in the ASL and process the sequences at different scales separately. We apply average pooling to downsample the input sequence $\boldsymbol{H} \in \mathbb{R}^{N \times L \times h}$ into $m$ different scales, resulting in a set of multi-scale spatiotemporal sequences $\mathcal{H} = \{\boldsymbol{H}^0, \cdots, \boldsymbol{H}^m\}$, where $\boldsymbol{H}^i \in \mathbb{R}^{N \times \lfloor \frac{L}{2^i} \rfloor \times h}$, and $i \in \{0, \cdots, m\}$. The finest scale sequence $\boldsymbol{H}^0$ corresponds to the original input and captures the most detailed temporal information. The coarsest scale sequence $\boldsymbol{H}^m$ contains the macro-level changes.

Next, we apply a moving average to smooth periodic fluctuations and extract both trend and seasonal components. For the input sequence $\boldsymbol{H}^i$, the decomposition proceeds are:

$$\boldsymbol{H}^i_T = AvgPool(Padding(\boldsymbol{H}^i)), \quad \boldsymbol{H}^i_S = \boldsymbol{H}^i - \boldsymbol{H}^i_T, \tag{4}$$

where $\boldsymbol{H}^i_T, \boldsymbol{H}^i_S \in \mathbb{R}^{N \times \lfloor \frac{L}{2^i} \rfloor \times h}$ denote the extracted trend and seasonal components, respectively. We apply $AvgPool(\cdot)$ to perform a moving average. $Padding(\cdot)$ refers to the process of copying the last time step of the sequence to preserve the original sequence length after decomposition. In the following modules, we will process the temporal features extracted from the trend and seasonal components separately.

For the trend component $\boldsymbol{H}^i_T$, we use $TCN$ (Fan et al., 2023) to extract trend features, as follows:

$$\hat{\boldsymbol{H}}^i_T = TCN(\boldsymbol{H}^i_T), \tag{5}$$

where $\hat{\boldsymbol{H}}^i_T \in \mathbb{R}^{N \times \lfloor \frac{L}{2^i} \rfloor \times h}$ represents the trend features, and $TCN(\cdot)$ is composed of two stacked dilated convolution layers (Luo & Wang, 2024).

Inspired by Yi et al. (2025), we devise the Season Learner module to extract season features, which adopts a frequency-domain approach. Specifically, for the seasonal component $\boldsymbol{H}^i_S$, we have:

$$\boldsymbol{R}^i, \boldsymbol{I}^i = FFT(\boldsymbol{H}^i_S), \quad \hat{\boldsymbol{H}}^i_S = iFFT(Linear(\boldsymbol{R}^i), Linear(\boldsymbol{I}^i)), \tag{6}$$

where $FFT(\cdot)$ denotes the Fourier transform. $\boldsymbol{R}^i, \boldsymbol{I}^i \in \mathbb{R}^{N \times d \times h}$ represent the real and imaginary parts of the frequency components, respectively, with $d$ being the number of frequency components. $iFFT(\cdot)$ denotes the inverse Fourier transform. $\hat{\boldsymbol{H}}^i_S \in \mathbb{R}^{N \times \lfloor \frac{L}{2^i} \rfloor \times h}$ represents the seasonal features.

Subsequently, we will construct dynamic adjacency matrices for different patterns and time scales based on these features. The process is as follows:

$$\tilde{\boldsymbol{A}}^i = \sigma(\beta_1 (\sum_l \hat{\boldsymbol{H}}^i_{:,l,:}) \boldsymbol{M}^{i\top} + \beta_2 (\sum_l \hat{\boldsymbol{H}}^i_{:,l,:})(\sum_l \hat{\boldsymbol{H}}^i_{:,l,:})^\top), \tag{7}$$

where the input feature $\hat{\boldsymbol{H}}^i \in \mathbb{R}^{N \times \lfloor \frac{L}{2^i} \rfloor \times h}$ represents either $\hat{\boldsymbol{H}}^i_S$ or $\hat{\boldsymbol{H}}^i_T$, $\boldsymbol{M}^i \in \mathbb{R}^{N \times h}$, $\beta_1 \in \mathbb{R}$, and $\beta_2 \in \mathbb{R}$ are trainable parameters. Thus, the adjacency matrix $\tilde{\boldsymbol{A}}^i \in \mathbb{R}^{N \times N}$ represents the implicit relationships between nodes. When the input feature is seasonal, $\tilde{\boldsymbol{A}}^i$ represents seasonal relationships, denoted $\tilde{\boldsymbol{A}}^i_S$; when the input feature is trend, $\tilde{\boldsymbol{A}}^i$ represents trend relationships, denoted $\tilde{\boldsymbol{A}}^i_T$.

By extracting inter-node relationships, we obtain $\boldsymbol{A}_L, \tilde{\boldsymbol{A}}^i_S, \tilde{\boldsymbol{A}}^i_T$, which represent dynamic relationships, seasonal relationships, and trend relationships among nodes, respectively. Following that, we leverage GCN (Kipf & Welling, 2017) and these adjacency matrices to aggregate information:

$$\boldsymbol{Z}^i_T = Linear([GCN(\boldsymbol{H}^i_T, \tilde{\boldsymbol{A}}^i_T), GCN(\boldsymbol{H}^i, \boldsymbol{A}_L)]), \tag{8}$$

$$\boldsymbol{Z}^i_S = Linear([GCN(\boldsymbol{H}^i_S, \tilde{\boldsymbol{A}}^i_S), GCN(\boldsymbol{H}^i, \boldsymbol{A}_L)]), \tag{9}$$

where $\boldsymbol{Z}^i_T, \boldsymbol{Z}^i_S \in \mathbb{R}^{N \times \lfloor \frac{L}{2^i} \rfloor \times h}$ represent the aggregated information based on different adjacency matrices. $GCN(\boldsymbol{H}, \boldsymbol{A}) := \sigma(\hat{\boldsymbol{A}} \boldsymbol{H} \boldsymbol{W})$, where $\boldsymbol{W} \in \mathbb{R}^{h \times h}$ is a learnable parameter, $\boldsymbol{H}$ is one of $\hat{\boldsymbol{H}}^i$, $\hat{\boldsymbol{H}}^i_S$, or $\hat{\boldsymbol{H}}^i_T$, and $\hat{\boldsymbol{A}} = \tilde{\boldsymbol{D}}^{-\frac{1}{2}}(\boldsymbol{A} + \boldsymbol{I}_N)\tilde{\boldsymbol{D}}^{-\frac{1}{2}} \in \mathbb{R}^{N \times N}$, with $\tilde{\boldsymbol{D}}_{i,i} := \sum_j (\boldsymbol{A} + \boldsymbol{I}_N)_{i,j}$, and $\boldsymbol{I}_N \in \mathbb{R}^{N \times N}$ denoting the identity matrix. $[\cdot, \cdot]$ denotes feature-wise concatenation, and $Linear(\cdot)$ projects the concatenated feature of dimension $2h$ back to $h$.

In the following, we adopt multi-scale fusion strategies (Wang et al., 2024) for trend and seasonal features. For seasonal features, we use a bottom-up approach to progressively aggregate information

from finer scales to coarser scales. This strategy introduces detailed information into coarse-scale seasonal modeling. Specifically, the seasonal component fusion process is:

$$\text{for } i \text{ in } 1 \longrightarrow m \text{ do: } \quad \boldsymbol{Z}_S^i = \boldsymbol{Z}_S^i + Linear(\boldsymbol{Z}_S^{i-1}), \tag{10}$$

$Linear(\cdot)$ adjusts the temporal length, with input dimension $\left\lfloor \frac{L}{2^{i-1}} \right\rfloor$ and output dimension $\left\lfloor \frac{L}{2^i} \right\rfloor$.

In contrast to seasonal features, detailed information on trend features may introduce noise when modeling macro trends. On the other hand, coarse-scale time series may provide macro-level information for finer-scale sequences. Thus, we adopt a top-down fusion approach, using macro-level knowledge from coarser scales to guide finer-scale trend modeling. Specifically, the trend features fusion process is:

$$\text{for } i \text{ in } (m-1) \longrightarrow 0 \text{ do: } \quad \boldsymbol{Z}_T^i = \boldsymbol{Z}_T^i + Linear(\boldsymbol{Z}_T^{i+1}), \tag{11}$$

where the input and output dimension of $Linear(\cdot)$ are $\left\lfloor \frac{L}{2^{i+1}} \right\rfloor$ and $\left\lfloor \frac{L}{2^i} \right\rfloor$, respectively. After multi-scale fusion, the final output of the ASL is $\boldsymbol{Z}^i = \boldsymbol{Z}_T^i + \boldsymbol{Z}_S^i$, where $i = [1, \ldots, m]$.

## 3.5 OUTPUT LAYER

After processing with ASL, we obtain multi-scale spatiotemporal features $\boldsymbol{Z}^i \in \mathbb{R}^{N \times \left\lfloor \frac{L}{2^i} \right\rfloor \times h}$. To fully utilize these features, we aggregate the predictions from all scales. The process is as follows:

$$\hat{\boldsymbol{\mathcal{X}}}^i = FFD_i(\boldsymbol{Z}^i), \text{ where } i \in (0, \cdots, m), \quad \hat{\boldsymbol{\mathcal{X}}} = \sum_{i=0}^m \hat{\boldsymbol{\mathcal{X}}}^i, \tag{12}$$

where $\hat{\boldsymbol{\mathcal{X}}}^i \in \mathbb{R}^{N \times O \times C}$ represents the prediction for each scale, while $\hat{\boldsymbol{\mathcal{X}}} \in \mathbb{R}^{N \times O \times C}$ is the final output. $FFD_i(\cdot)$ uses two linear layers without activation functions, first mapping the input sequence length from $\left\lfloor \frac{L}{2^i} \right\rfloor$ to the prediction length $O$, then projecting the hidden features into $C$ variables. A comparative discussion of DPGNet with existing graph generation methods can be found in A.1.

Table 1: Results of regular-term spatiotemporal prediction.

| Dataset | Model | $O = 3$ | | $O = 6$ | | $O = 12$ | |
|---|---|---|---|---|---|---|---|
| | | MAE | MSE | MAE | MSE | MAE | MSE |
| METR-LA | PGCN (Shin & Yoon, 2024) | 3.5403 | 65.8224 | 4.5256 | 93.8632 | 5.6879 | 137.4897 |
| | PMC-GCN (Gao et al., 2024) | 3.5768 | 66.5595 | 4.5233 | 92.7042 | 5.8151 | 139.9323 |
| | STIDGCN (Liu & Zhang, 2024) | 3.7565 | 64.9475 | 4.3655 | 92.5482 | 5.4594 | 135.1360 |
| | TESTAM (Lee & Ko, 2024) | 3.7806 | 69.9582 | 4.6363 | 98.6943 | 6.1086 | 145.6487 |
| | WAVGCRN (Qian & Mallick, 2024) | 4.9907 | 82.6117 | 5.3720 | 111.7545 | 6.9338 | 160.7621 |
| | DPGNet | **3.3575** | **64.7494** | **4.1814** | **90.4401** | **5.3846** | **131.3522** |
| PEMS-Bay | PGCN (Shin & Yoon, 2024) | 1.1373 | 5.2974 | 1.4082 | 8.9054 | 1.7217 | 13.5844 |
| | PMC-GCN (Gao et al., 2024) | 1.1580 | 5.4770 | 1.4937 | 9.7375 | 2.3516 | 24.1907 |
| | STIDGCN (Liu & Zhang, 2024) | 1.1292 | 5.1683 | 1.3769 | 8.4827 | 1.6606 | 12.7549 |
| | TESTAM (Lee & Ko, 2024) | 1.1930 | 5.7653 | 1.5325 | 10.2310 | 1.8279 | 14.7671 |
| | WAVGCRN (Qian & Mallick, 2024) | 1.3138 | 6.2585 | 1.6025 | 10.0959 | 2.2676 | 18.1493 |
| | DPGNet | **1.1101** | **5.0625** | **1.3600** | **8.2639** | **1.6499** | **12.3299** |
| PEMS08 | PGCN (Shin & Yoon, 2024) | 15.7695 | 820.0950 | 17.9700 | 1197.8729 | 20.8111 | 1709.7067 |
| | PMC-GCN (Gao et al., 2024) | 15.9641 | 840.1476 | 18.3765 | 1205.8395 | 22.0897 | 1808.6223 |
| | STIDGCN (Liu & Zhang, 2024) | 15.3111 | **790.1665** | 17.6426 | 1157.6870 | 19.9215 | 1651.0813 |
| | TESTAM (Lee & Ko, 2024) | 29.6833 | 1869.7495 | 23.1501 | 1782.7395 | 30.8176 | 2511.6235 |
| | WAVGCRN (Qian & Mallick, 2024) | 22.6586 | 1314.9834 | 21.0412 | 1359.0946 | 28.9435 | 2339.8214 |
| | DPGNet | **15.2474** | 790.1721 | **17.3430** | **1157.3604** | **19.5711** | **1627.3156** |
| Electricity | PGCN (Shin & Yoon, 2024) | 0.2766 | 0.1774 | 0.2811 | 0.1811 | 0.2859 | 0.1859 |
| | PMC-GCN (Gao et al., 2024) | **0.1523** | 0.0600 | 0.1752 | 0.0770 | **0.1808** | 0.0870 |
| | STIDGCN (Liu & Zhang, 2024) | 0.1542 | 0.0585 | 0.1706 | 0.0711 | 0.1832 | 0.0830 |
| | TESTAM (Lee & Ko, 2024) | 0.1838 | 0.0808 | 0.1865 | 0.0840 | 0.2029 | 0.1035 |
| | WAVGCRN (Qian & Mallick, 2024) | - | - | - | - | - | - |
| | DPGNet | 0.1587 | **0.0584** | **0.1646** | **0.0672** | 0.1821 | **0.0801** |
| Weather | PGCN (Shin & Yoon, 2024) | 0.3211 | 0.2679 | 0.3230 | 0.2763 | 0.3352 | 0.2735 |
| | PMC-GCN (Gao et al., 2024) | **0.0676** | **0.0443** | 0.0904 | 0.0582 | 0.1169 | 0.0756 |
| | STIDGCN (Liu & Zhang, 2024) | 0.0743 | 0.0462 | 0.1646 | 0.0672 | 0.1225 | 0.0758 |
| | TESTAM (Lee & Ko, 2024) | 0.1047 | 0.0565 | 0.1350 | 0.0779 | 0.1479 | 0.0903 |
| | WAVGCRN (Qian & Mallick, 2024) | 0.0826 | 0.0565 | 0.0949 | 0.0590 | 0.1441 | 0.0830 |
| | DPGNet | 0.0690 | 0.0445 | **0.0841** | **0.0564** | 0.1095 | 0.0742 |

## 4 EXPERIMENTS

In this section, we assess the performance of DPGNet using five real-world benchmark datasets. Additionally, we conduct comprehensive analytical experiments to evaluate the effectiveness of each module and examine the computational cost. In all prediction results in the following, **bolded values** represent the best results, underlined values represent the second-best, and a dash ("-") indicates models that failed to train due to out-of-memory issues.

Table 2: Results of long-term spatiotemporal prediction.

| Dataset | Model | O = 24 | | O = 36 | | O = 48 | |
|---|---|---|---|---|---|---|---|
| | | MAE | MSE | MAE | MSE | MAE | MSE |
| METR-LA | PGCN (Shin & Yoon, 2024) | 7.3111 | 194.2460 | 8.3067 | 237.6264 | 9.1226 | 262.8521 |
| | PMC-GCN (Gao et al., 2024) | 7.4773 | 195.8361 | 8.9720 | 239.7505 | 9.3148 | 267.5599 |
| | STIDGCN (Liu & Zhang, 2024) | 7.5140 | 197.2141 | 8.6893 | 235.4212 | 9.9960 | 278.8592 |
| | TESTAM (Lee & Ko, 2024) | 7.5162 | 204.7470 | 8.9428 | 257.9806 | 9.4628 | 274.6246 |
| | WAVGCRN (Qian & Mallick, 2024) | 16.7661 | 509.1225 | 10.6540 | 283.3590 | 18.5267 | 541.1127 |
| | DPG-Net | **7.1034** | **191.6066** | **7.8676** | **223.1617** | **8.6342** | **249.3086** |
| PEMS08 | PGCN (Shin & Yoon, 2024) | 24.2191 | 2279.4170 | 25.6723 | 2611.3066 | 30.8520 | 3160.2815 |
| | PMC-GCN (Gao et al., 2024) | 26.4791 | 2530.5159 | 29.1199 | 3047.9412 | 30.1908 | 3196.2134 |
| | STIDGCN (Liu & Zhang, 2024) | 23.8807 | 2240.3438 | 25.4253 | 2526.1126 | 26.6498 | 2718.4949 |
| | TESTAM (Lee & Ko, 2024) | 37.1592 | 3446.3430 | 42.3845 | 4308.4917 | 48.4166 | 5585.1382 |
| | WAVGCRN (Qian & Mallick, 2024) | 34.5698 | 3212.1919 | 44.2962 | 4718.5728 | 49.0788 | 5506.2710 |
| | DPG-Net | **23.0820** | **2052.7918** | **23.5626** | **2268.9053** | **24.6720** | **2463.2017** |

## 4.1 EXPERIMENTAL SETUP

**Datasets:** We conduct experiments on five widely used real-world datasets: METR-LA (Li et al., 2018), PEMS08 (Guo et al., 2021), PEMS-Bay (Li et al., 2018), Electricity (Lai et al., 2018), and Weather (Wu et al., 2021). The detailed statistical properties of these datasets and their initial adjacency configuration are provided in A.2.

**Baselines:** In this work, we select five recent baseline models: PGCN (Shin & Yoon, 2024), PMC-GCN (Gao et al., 2024), STIDGCN (Liu & Zhang, 2024), TESTAM (Lee & Ko, 2024), and WAVGCRN (Qian & Mallick, 2024). A detailed introduction of them is provided in the appendix A.3

**Metrics:** We adopt two commonly used metrics in spatio-temporal forecasting, Mean Absolute Error (MAE) and Mean Squared Error (MSE). We calculated the prediction errors for all prediction points, unlike TESTAM (Lee & Ko, 2024), which only computed the prediction errors for non-zero points. The detailed formulations are provided in A.4

**Hyperparameter Settings :** In our experiments, all baseline models are used with their optimal settings as reported in Shin & Yoon (2024); Gao et al. (2024); Liu & Zhang (2024); Lee & Ko (2024); Qian & Mallick (2024). For DPGNet, we set the hidden dimension $h = 64$. The model is trained with the Adam optimizer using MAE as the loss function. Training runs for 100 epochs with a batch size of 32. The learning rate is initialized at 0.0005 and follows a cosine decay schedule.

## 4.2 MAIN RESULTS

**Regular-Term Spatial Temporal Prediction:** To evaluate DPGNet's performance on spatiotemporal forecasting tasks, we conduct experiments on five commonly used datasets. For METR-LA, PEMS08, and PEMS-Bay, we set the input length $L$ to 12, with prediction lengths $O$ of 3, 6, and 12, corresponding to time horizons of 15, 30, and 60 minutes, respectively. For the Electricity and Weather datasets, we set $L$ to 168, with prediction lengths $O$ of 3, 6, and 12. For Weather, these correspond to 30, 60, and 120 minutes, while for Electricity, they represent 3, 6, and 12 hours. As shown in Table 1, DPGNet outperforms baseline models in the majority of forecasting scenarios. For example, on the METR-LA dataset, DPGNet achieves an average MAE reduction of 3.65% compared to the second-best model across all prediction horizons.

**Long-Term Spatial Temporal Prediction:** Benefiting from its multi-scale processing framework, DPGNet demonstrates strong performance in long-term spatiotemporal forecasting tasks. To validate this, we compare DPGNet's long-term forecasting performance against baseline models. For METR-LA and PEMS08, the input length $L$ is set to 12, and prediction lengths $O$ are set to 24, 36, and 48, corresponding to 120, 180, and 240 minutes. As shown in Table 2, DPGNet outperforms all baseline models. For example, on METR-LA, DPGNet achieves an average MAE reduction of 4.51% compared to the second-best model across all prediction horizons. On PEMS08, DPGNet reduces the MSE by an average of 9.31%. Notably, DPGNet's performance advantage increases with the prediction horizon, achieving the largest MAE reduction of 7.42% at the 240-minute horizon on PEMS08. At 120 and 180 minutes, the reductions are 3.34% and 7.33%, respectively. More experimental results are given in Appendix A.5.

Table 3: The replacement experiment results show performance improvement (shaded) after replacing the original graph generation method with AGL. 'O-Parameters' and 'A-Parameters' refer to the number of trainable parameters in the original graph generation modules and AGL, respectively.

| Dataset | Model | | 15 minutes | | 60 minutes | | O-Parameters | A-Parameters |
|---|---|---|---|---|---|---|---|---|
| | | | MAE | MSE | MAE | MSE | | |
| METR-LA | GWNet (Wu et al., 2019) | vanilla | 3.4800 | 66.8846 | 5.6984 | 134.0552 | 4140 | 5888 |
| | | w/ AGL | 3.3575(3.52%↓) | 64.7486(3.19%↓) | 5.3846(5.51%↓) | 131.3513(2.02%↓) | | |
| | PMC-GCN (Gao et al., 2024) | vanilla | 3.5768 | 66.5593 | 5.8151 | 139.9326 | 257094 | 2912 |
| | | w/ AGL | 3.5700(0.19%↓) | 67.0761(0.78%↑) | 5.7812(0.58%↓) | 136.1889(2.68%↓) | | |
| | STIDGCN (Liu & Zhang, 2024) | vanilla | 3.7565 | 64.9473 | 5.4594 | 135.1364 | 39753 | 2912 |
| | | w/ AGL | 3.5665(5.06%↓) | 64.4809(0.72%↓) | 5.4104(0.90%↓) | 131.5148(2.69%↓) | | |
| | STGCN (Yan et al., 2018) | vanilla | 3.6539 | 66.7832 | 5.6993 | 134.4466 | 0 | 2912 |
| | | w/ AGL | 3.5311(3.36%↓) | 66.2596(0.79%↓) | 5.9036(3.58%↑) | 138.7884(3.23%↑) | | |
| | WAVGCRN (Qian & Mallick, 2024) | vanilla | 4.9907 | 82.6108 | 6.9338 | 160.7631 | 0 | 2912 |
| | | w/ AGL | 3.6849(26.16%↓) | 67.4041(18.41%↓) | 6.7905(2.07%↓) | 149.5729(6.96%↓) | | |
| PEMS08 | GWNet (Wu et al., 2019) | vanilla | 15.8843 | 830.0247 | 21.1694 | 1745.7639 | 3400 | 5814 |
| | | w/ AGL | 15.5646(2.01%↓) | 826.3616(0.44%↓) | 20.8184(1.66%↓) | 1765.5440(1.13%↑) | | |
| | PMC-GCN (Gao et al., 2024) | vanilla | 15.9641 | 840.1459 | 22.0897 | 1808.6214 | 173400 | 3030 |
| | | w/ AGL | 15.6584(1.91%↓) | 817.4102(2.71%↓) | 21.6807(1.85%↓) | 1778.2541(1.68%↓) | | |
| | STIDGCN (Liu & Zhang, 2024) | vanilla | 15.3111 | 790.1659 | 19.8736 | 1641.5082 | 32649 | 3030 |
| | | w/ AGL | 15.2417(0.45%↓) | 786.2132(0.50%↓) | 19.8284(0.27%↓) | 1677.8730(2.22%↑) | | |
| | STGCN (Yan et al., 2018) | vanilla | 18.1283 | 951.7638 | 22.9451 | 1931.8759 | 0 | 3030 |
| | | w/ AGL | 18.1212(0.03%↓) | 921.6303(3.17%↓) | 23.1401(0.85%↑) | 1924.5240(0.38%↓) | | |
| | WAVGCRN (Qian & Mallick, 2024) | vanilla | 22.6586 | 1314.9849 | 28.9435 | 2339.8194 | 0 | 3030 |
| | | w/ AGL | 18.2564(19.43%↓) | 1002.9889(23.73%↓) | 20.3240(3.41%↓) | 1897.6740(18.90%↓) | | |

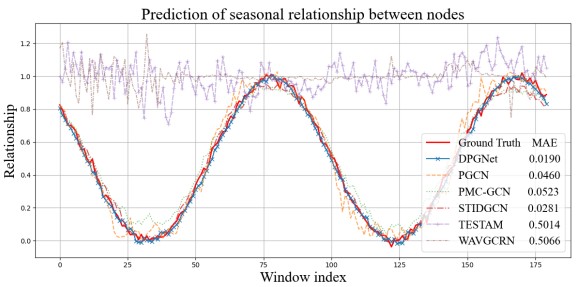

Figure 4: Prediction of seasonal relationship between nodes. The y-axis represents similarity, and the x-axis denotes the predictive window index. The red line indicates the ground-truth.

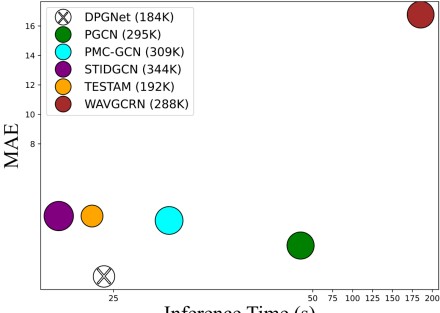

Figure 5: Computation burden comparison on METR-LA. Performance (y-axis), interference time (x-axis), and trainable parameters (size of the shape) of methods.

## 4.3 MODEL ANALYSIS

**AGL Analysis**: Our proposed AGL, as a plug-and-play graph generation module, can effectively capture dynamic relationships among nodes. To verify the advantages of the AGL, we design a module replacement experiment in this section. Specifically, for baseline models that include graph generation modules, we replace their original modules with the proposed AGL and compare the prediction accuracy before and after the replacement to evaluate its effectiveness. To ensure a fair comparison, we use the same hidden dimensions and hyperparameters across all experiments.

Table 3 presents the changes in prediction accuracy after replacing the graph generation modules in baseline models with AGL. GWNet, PMC-GCN, and STIDGCN learn their graphs via neural networks, whereas STGCN and WAVGCRN use only predefined adjacency matrices. It can be observed that AGL outperforms other methods in 85% of the scenarios. In addition, Table 3 compares the number of trainable parameters for AGL and other graph generation methods. Compared with other neural network-based methods, the AGL introduces fewer trainable parameters. For methods using predefined adjacency matrices, the AGL does not significantly increase the number of parameters. In summary, the AGL outperforms existing graph generation methods. For results with more prediction horizon settings, please refer to Appendix A.6.

**Season Learner Analysis**: In this experiment, we conduct two simulated nodes with predefined seasonal patterns and generate data to evaluate the ASL's ability to capture seasonal dependencies between nodes. For the generation process of the simulated data, please refer to Appendix A.7. Figure 4 shows the performance of different algorithms in capturing the seasonal relationships between

Table 4: Ablation study on METR-LA.

| Dateset | Model | 120 minutes | | 180 minutes | | 240 minutes | |
|---|---|---|---|---|---|---|---|
| | | MAE | MSE | MAE | MSE | MAE | MSE |
| METR-LA | DPGNet | **7.1034** | **191.6066** | **7.8676** | **223.1617** | **8.6342** | **249.3086** |
| | w/o season feature | 7.7387 | 204.8814 | 8.8438 | 235.1586 | 9.4877 | 255.3363 |
| | w/o trend feature | 7.6339 | 194.6127 | 8.6864 | 230.1645 | 9.2585 | 249.8750 |
| | w/o AGL | 8.0056 | 210.3098 | 9.1913 | 245.3591 | 9.9757 | 287.2614 |

simulated nodes. DPGNet, with an MAE of $0.0190$, significantly outperforms the others, highlighting its strong ability to capture seasonal dependencies across nodes.

**Ablation Study**: We conduct ablation studies to evaluate the contribution of DPGNet's core modules. Three variants are designed: w/o season feature: Remove seasonal feature processing from the ASL; w/o trend feature: Remove trend feature processing from the ASL; w/o AGL: Remove the AGL and replace its output $A_L$ with the original adjacency matrix $A$.

Table 4 presents the results of the ablation study on the METR-LA dataset. All core modules within DPGNet demonstrate their importance, as evidenced by a noticeable decline in prediction accuracy when any of these components is removed. Notably, removing the AGL results in performance drops of up to $12.70\%$ in MAE and $9.76\%$ in MSE, highlighting its critical role in enhancing DPGNet's accuracy. Additional ablation study results can be found in Appendix A.8. For the sensitivity analysis of the multi-scale parameter $m$, please see Appendix A.9.

**Efficiency Analysis**: We will compare and analyze the efficiency of DPGNet and several baseline models using the METR-LA dataset. To ensure robustness and reliability, each experiment is repeated five times, and the average results are reported. In this setting, the input sequence length $L$ is set to $12$, and the prediction horizon $O$ to $24$. Specifically, we evaluate and compare prediction accuracy, inference time, and number of trainable parameters in DPGNet, PGCN, PMC-GCN, STIDGCN, TESTAM, and WAVGCRN. All models are tested under the same experimental environment, with a fixed batch size of $64$. As shown in Figure 5, DPGNet achieves the highest prediction accuracy while maintaining the lowest number of trainable parameters (184K). In terms of inference time, DPGNet ranks just behind STIDGCN and TESTAM, demonstrating a favorable balance between performance and efficiency.

## 5 CONCLUSION

In this paper, we propose DPGNet, a novel spatiotemporal prediction model that effectively captures both dynamic relationships between nodes and complex temporal patterns in spatiotemporal data. DPGNet introduces a plug-and-play graph generation module, the AGL, which combines the GRU's gating mechanism with self-attention to dynamically learn implicit inter-node dependencies. To better model temporal patterns, DPGNet also incorporates the ASL, which integrates temporal decomposition, multi-scale processing, and pattern-specific graph construction strategies. These components jointly enhance the model's capability to uncover complex temporal patterns and deliver more reliable long-term forecasts. We evaluate DPGNet on five benchmark datasets. Compared to state-of-the-art models, DPGNet achieves higher prediction accuracy, fewer trainable parameters, and competitive inference speed. Further analysis shows that the AGL outperforms existing graph generation methods and can be easily applied to improve other models.

In future work, we aim to develop an automatic selection mechanism for the multi-scale hyperparameter $m$ in ASL and extend DPGNet to tasks without predefined graphs, enabling adaptive learning of implicit spatial dependencies.

### ETHICS STATEMENT

This study complies with all relevant ethical and legal standards, does not involve human subjects, and uses publicly available datasets that meet privacy and security requirements. We have ensured fairness, avoided biases, and have no conflicts of interest or funding that could influence the results. All research ethics guidelines have been followed, maintaining the highest standards of integrity.

REPRODUCIBILITY STATEMENT

Anonymously downloadable source code for the proposed algorithm is provided.

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
