# A APPENDIX

This appendix presents extended experimental results and methodological details to complement the main body of the paper.

## A.1 DISCUSSION ON DIFFERENCES WITH OTHER GRAPH GENERATION METHODS

In existing spatiotemporal prediction models, graph generation methods can be divided into two categories: (1) Prior Graphs: These are graph structures known to the model in advance, typically static and representing physical connections between nodes. (2) Dynamic Graphs: These are graph structures learned by the model based on input spatiotemporal data. Existing methods for generating dynamic graphs can be further divided into two types: Learnable Parameters: In models like GWNet (Wu et al., 2019) and PGCN (Shin & Yoon, 2024), learnable parameters (denoted as $A \in \mathbb{R}^{N \times N}$) are set as the adjacency matrix and updated through backpropagation. However, since $A$ cannot be adjusted based on test data during inference, it remains essentially static. Module Generation: In STIDGCN (Liu & Zhang, 2024), dynamic graphs are generated through a pattern bank that integrates attention mechanisms. Meanwhile, MegaCRN (Jiang et al., 2023) generates dynamic graphs through memory cells and updates them based on input data.

DPGNet also generates dynamic graphs through a modular generation approach. Unlike existing methods, DPGNet's AGL leverages G-RNN to generate corresponding local dynamic graphs for each patch. Through a recursive process, the generation of the current dynamic graph considers information from the previous patch, allowing the model to retain important node information while discarding weak connections. As a result, the dynamic graphs generated by DPGNet through AGL can sense local changes in spatial relationships, enabling the model to promptly adjust the learned adjacency relationships. Additionally, these dynamic graphs are less affected by noise, enhancing the model's ability to focus on relevant spatiotemporal patterns.

## A.2 DATASET

Table 5: Statistics of datasets. METR-LA, PEMS-Bay, and PEMS08 are widely used spatiotemporal series datasets, while Electricity and Weather are commonly used time series datasets.

| Type | Dataset | Node | Edges | Time Step | Time Interval | Time Range |
|---|---|---|---|---|---|---|
| Speed | METR-LA | 207 | 1722 | 34272 | 5 minutes | Mar 1st 2012 to Jun 30th 2012 |
| Flow | PEMS-Bay | 325 | 2694 | 52116 | 5 minutes | Jan 1st 2017 to June 30th 2017 |
| | PEMS08 | 170 | 548 | 17856 | 5 minutes | July 1st 2018 to Aug 31th 2018 |
| Time-Series | Electricity | 321 | - | 26304 | 1 hour | July 1st 2016 to July 2nd 2019 |
| | Weather | 21 | - | 52695 | 10 minutes | Jan 1st 2020 to Jan 1st 2021 |

In this section, we present the statistical properties and the initial adjacency configuration of the datasets used in the paper for spatiotemporal prediction tasks. Table 5 presents the statistical properties of datasets. For PEMS-Bay, METR-LA, and PEMS08, we follow previous studies Lee & Ko (2024); Bai et al. (2020); Liu & Zhang (2024) and use the physical connectivity among nodes as the initial adjacency configuration. For Weather and Electricity, no explicit relationships among nodes are provided. Therefore, we use the identity matrix as the initial adjacency configuration.

## A.3 BASELINE MODELS

We conduct comparative experiments with five state-of-the-art spatialtemporal prediction models. Below is a brief introduction to each of these models.

- PGCN (Shin & Yoon, 2024) proposes a novel approach that enables the dynamic updating of graph structures during the training phase to capture complex, time-varying spatial correlations in transportation networks and adapt to online input data across both training and testing stages.

- PMC-GCN (Gao et al., 2024) introduces an innovative personalized-enhanced graph convolution that employs learnable diagonal matrices combined with multi-head self-attention modules to improve prediction robustness.

- STIDGCN (Liu & Zhang, 2024) designs a dynamic interactive learning framework, comprising spatial and temporal modules, to effectively capture spatiotemporal dependencies in traffic flow data.

- TESTAM (Lee & Ko, 2024) presents a pioneering hybrid model that integrates three specialized experts and incorporates time-aware attention mechanisms along with dynamic graph modeling to address both recurrent and non-recurrent traffic patterns.

- WAVGCRN (Qian & Mallick, 2024) develops a multi-scale spatiotemporal prediction framework that integrates wavelet transforms with graph convolutional recurrent networks. The model synergizes prior road network knowledge with data-driven graph learning to achieve dynamic graph structure adaptation, thereby enhancing spatiotemporal dependency modeling and forecasting performance.

## A.4 Evaluation Metrics

The calculation formula for MAE and MSE used in the experiment is as follows: Let the predictions be denoted as: $\hat{\mathcal{X}} = (\hat{\boldsymbol{X}}_{L+1}, \hat{\boldsymbol{X}}_{L+2}, \ldots, \hat{\boldsymbol{X}}_{L+O}) \in \mathbb{R}^{N \times O \times C}$. For the time step $t \in \{L+1, \ldots, L+O\}$, the corresponding ground truth and prediction are respectively expressed as $\boldsymbol{X}_t = (\boldsymbol{x}_{t,1}, \boldsymbol{x}_{t,2}, \ldots, \boldsymbol{x}_{t,N}) \in \mathbb{R}^{C \times N}$ and $\hat{\boldsymbol{X}}_t = (\hat{\boldsymbol{x}}_{t,1}, \hat{\boldsymbol{x}}_{t,2}, \ldots, \hat{\boldsymbol{x}}_{t,N}) \in \mathbb{R}^{C \times N}$, where $\boldsymbol{x}_{t,n} \in \mathbb{R}^C$ denotes the feature of node $n$ at time step $t$, and $C$ represents the feature dimension, and $\hat{\boldsymbol{x}}_{t,n} \in \mathbb{R}^C$ denotes the corresponding prediction. The formulations for Mean Absolute Error (MAE) and Mean Squared Error (MSE) are respectively defined as:

$$\text{MAE} := \frac{1}{O} \frac{1}{N} \sum_{t=L+1}^{L+O} \sum_{n=1}^{N} \frac{1}{C} \left\| \boldsymbol{x}_{t,n} - \hat{\boldsymbol{x}}_{t,n} \right\|_1,$$

$$\text{MSE} := \frac{1}{O} \frac{1}{N} \sum_{t=L+1}^{L+O} \sum_{n=1}^{N} \frac{1}{C} \left\| \boldsymbol{x}_{t,n} - \hat{\boldsymbol{x}}_{t,n} \right\|_2^2,$$

where, $\| \cdot \|_1$ and $\| \cdot \|_2$ denote the $\ell_1$ and $\ell_2$ norms, respectively. These formulations express the errors as the average norm of the discrepancy vectors across all time steps and nodes.

## A.5 Long-Term Spatial Temporal Prediction

In this section, we will present the complete experimental results to highlight the accuracy advantage of DPGNet in long-term spatial-temporal prediction tasks. Table 6 presents the complete results of DPGNet on the long-term spatiotemporal prediction task. As shown in the Table, DPGNet achieved optimal results in 79.17% of the long-term prediction tasks and suboptimal results in the remaining predictions.

## A.6 AGL Analysis

In this section, we will present the complete experimental results to validate the effectiveness of the AGL module. Table 7 presents the adjacency matrices learned by the baseline models and DPGNet on the METR-LA and PEMS08 datasets. For PMC-GCN and WAVGCRN, they only account for explicit relationships between nodes, without considering implicit relationships. Therefore, we did not include the adjacency matrices learned by these models in Table 7. As shown in the table, the adjacency matrix learned by DPGNet exhibits the fewest weak connections and most connections that contain significant information. Table 8 presents the complete results of the replacement experiment. For an introduction to the replacement experiment, please refer to Section 4.3. It can be observed that in 81.25% of the experimental scenarios, AGL outperforms the existing graph generation methods.

Table 6: Results of long-term spatiotemporal prediction. We report the average value and standard deviation of experiments. **Bolded values** indicate the best results, underlined values indicate the second-best results.

| Dataset | Model | $O = 24$ | | $O = 36$ | | $O = 48$ | |
|---|---|---|---|---|---|---|---|
| | | MAE | MSE | MAE | MSE | MAE | MSE |
| METR-LA | PGCN | 7.3111 ±0.1075 | 194.2460 ±1.8733 | 8.3067 ±0.1487 | 237.6264 ±0.6319 | 9.1226 ±0.4191 | 262.8521 ±2.8221 |
| | PMC-GCN | 7.4773 ±0.1788 | 195.8361 ±1.8718 | 8.9720 ±0.0865 | 239.7505 ±1.9711 | 9.3148 ±0.1417 | 267.5599 ±1.9818 |
| | STIDGCN | 7.5140 ±0.1747 | 197.2141 ±1.7023 | 8.6893 ±0.1695 | 235.4212 ±2.2484 | 9.9960 ±0.0375 | 278.8592 ±5.1389 |
| | TESTAM | 7.5162 ±0.2481 | 204.7470 ±7.2873 | 8.9428 ±0.2868 | 257.9806 ±6.7583 | 9.4628 ±0.1818 | 274.6246 ±5.1078 |
| | WAVGCRN | 16.7661 ±0.3883 | 509.1225 ±10.5507 | 10.6540 ±1.0256 | 283.3590 ±27.5473 | 18.5267 ±1.8289 | 541.1127 ±87.5866 |
| | DPGNet | **7.1034** ±0.2800 | **191.6066** ±4.4801 | **7.8676** ±0.4021 | **223.1617** ±5.2667 | **8.6342** ±0.3057 | **249.3086** ±2.6067 |
| PEMS08 | PGCN | 24.2191 ±0.3345 | 2279.4170 ±21.5158 | 25.6723 ±0.5522 | 2611.3066 ±45.6822 | 30.8520 ±1.4169 | 3160.2815 ±129.2806 |
| | PMC-GCN | 26.4791 ±0.3335 | 2530.5159 ±34.0195 | 29.1199 ±0.1475 | 3047.9412 ±13.5515 | 30.1908 ±0.3271 | 3196.2134 ±25.2430 |
| | STIDGCN | 23.8807 ±0.0020 | 2240.3438 ±0.7330 | 25.4253 ±0.0013 | 2526.1126 ±0.1567 | 26.6498 ±0.0017 | 2718.4949 ±0.6759 |
| | TESTAM | 37.1592 ±3.5786 | 3446.3430 ±446.6575 | 42.3845 ±3.1031 | 4308.4917 ±422.9279 | 48.4166 ±4.1474 | 5585.1382 ±622.0002 |
| | WAVGCRN | 34.5698 ±0.9612 | 3212.1919 ±123.5820 | 44.2962 ±1.0149 | 4718.5728 ±161.8669 | 49.0788 ±1.8447 | 5506.2710 ±249.7399 |
| | DPGNet | **23.0820** ±0.4422 | **2052.7918** ±33.791 | **23.5623** ±0.0505 | **2268.9053** ±5.3981 | **24.6720** ±0.1992 | **2463.2017** ±16.2884 |
| PEMS-Bay | PGCN | 2.0592 ±0.0296 | 18.7733 ±0.4506 | 2.2667 ±0.0128 | 21.9986 ±0.2535 | 2.4562 ±0.0331 | 24.5206 ±0.4006 |
| | PMC-GCN | 2.3946 ±0.0543 | 24.4524 ±0.2096 | 2.6632 ±0.0304 | 29.6252 ±0.3363 | 2.9132 ±0.0329 | 34.0172 ±0.3319 |
| | STIDGCN | 1.9394 ±0.0164 | **16.8079** ±0.1429 | **2.0612** ±0.01262 | **19.1240** ±0.1689 | 2.2206 ±0.0103 | 21.0813 ±0.0648 |
| | TESTAM | 2.2037 ±0.1150 | 20.8823 ±0.9980 | 2.4821 ±0.0833 | 25.0493 ±0.9822 | 2.7603 ±0.0908 | 29.6056 ±1.3966 |
| | WAVGCRN | 2.6232 ±0.1028 | 27.6689 ±1.1057 | 3.7462 ±0.0162 | 42.1594 ±0.9214 | 4.2947 ±0.1673 | 54.7341 ±3.0691 |
| | DPGNet | **1.9350** ±0.0077 | 17.2711 ±0.1273 | 2.0806 ±0.01243 | 19.3558 ±0.2204 | **2.1948** ±0.0100 | **20.9597** ±0.1162 |
| Weather | PGCN | 0.4522 ±0.0283 | 0.4086 ±0.0329 | 0.4924 ±0.0400 | 0.4641 ±0.0727 | 0.5955 ±0.0830 | 0.6402 ±0.1982 |
| | PMC-GCN | **0.1409** ±0.0139 | **0.0969** ±0.0037 | 0.1879 ±0.0033 | 0.1231 ±0.0014 | 0.2198 ±0.0209 | 0.1437 ±0.0185 |
| | STIDGCN | 0.1565 ±0.0026 | 0.1011 ±0.0008 | 0.1878 ±0.0198 | 0.1229 ±0.0158 | 0.2137 ±0.0019 | 0.1390 ±0.0008 |
| | TESTAM | 0.2145 ±0.0286 | 0.1445 ±0.0133 | 0.2452 ±0.0585 | 0.1693 ±0.0777 | 0.3082 ±0.0381 | 0.2938 ±0.0552 |
| | WAVGCRN | 0.1971 ±0.0039 | 0.1242 ±0.0027 | 0.2285 ±0.0103 | 0.1588 ±0.0112 | 0.2932 ±0.0255 | 0.2238 ±0.0288 |
| | DPGNet | 0.1489 ±0.0056 | 0.0988 ±0.0018 | **0.1771** ±0.0183 | **0.1177** ±0.0164 | **0.1990** ±0.0280 | **0.1327** ±0.0476 |

Table 7: The learned adjacency matrix of the model. Adjacency values within the range of $(0, 0.1]$ are considered weak connections.

| Datasets | Model | Similarity Range | | | |
|---|---|---|---|---|---|
| | | 0 | (0-0.1] | (0.1-0.5] | (0.5-1] |
| METR-LA | DPGNet | 90.94% | 7.47% | 1.31% | 0.28% |
| | GWNet | 0.00% | 98.84% | 1.07% | 0.09% |
| | PGCN | 0.00% | 100.00% | 0.00% | 0.00% |
| | STIDGCN | 20.29% | 79.71% | 0.00% | 0.00% |
| | TESTAM | 0.00% | 100.00% | 0.00% | 0.00% |
| PEMS08 | DPGNet | 89.76% | 8.95% | 0.93% | 0.37% |
| | GWNet | 0.00% | 99.03% | 0.86% | 0.11% |
| | PGCN | 0.00% | 100.00% | 0.00% | 0.00% |
| | STIDGCN | 10.14% | 89.86% | 0.00% | 0.00% |
| | TESTAM | 0.00% | 100.00% | 0.00% | 0.00% |

## A.7 SEASON LEARNER ANALYSIS

In this section, we introduce the generation process of the simulated data used in Section 4.3 and provide a detailed description of the experimental setup. In the experiment, we set both the historical window length $L$ and the prediction length $O$ to 96, and generate 900 data windows. For the $i$-th

Table 8: Results of the replacement experiment, with shading indicating performance improvement after replacing the original graph generation method with AGL.

| Dataset | Model | | 15 minutes | | 30 minutes | | 60 minutes | | 120 minutes | |
|---|---|---|---|---|---|---|---|---|---|---|
| | | | MAE | MSE | MAE | MSE | MAE | MSE | MAE | MSE |
| METR-LA | GWNet | vanilla | 3.4800 | 66.8846 | 4.4492 | 110.7876 | 5.6984 | 134.0552 | 7.3338 | 189.2081 |
| | | w/ AGL | 3.3575(3.52%↓) | 64.7486(3.19%↓) | 4.1814(6.02%↓) | 98.6049(11.00%↓) | 5.3846(5.51%↓) | 131.3513(2.02%↓) | 7.1034(3.14%↓) | 191.6066(1.21%↑) |
| | PMC-GCN | vanilla | 3.5768 | 66.5593 | 4.5233 | 92.7049 | 5.8151 | 139.9326 | 7.4774 | 195.8361 |
| | | w/ AGL | 3.5700(0.19%↓) | 67.0761(0.78%↑) | 4.6437(2.66%↑) | 93.7024(1.08%↑) | 5.7812(0.58%↓) | 136.1889(2.68%↓) | 7.9444(6.25%↑) | 206.2663(5.33%↑) |
| | STIDGCN | vanilla | 3.7565 | 64.9473 | 4.3655 | 92.5479 | 5.4594 | 135.1364 | 7.5140 | 197.2141 |
| | | w/ AGL | 3.5665(5.06%↓) | 64.4809(0.72%↓) | 4.3366(0.66%↓) | 91.3041(1.34%↓) | 5.4104(0.90%↓) | 131.5148(2.69%↓) | 6.9571(7.41%↓) | 184.8884(6.25%↓) |
| | STGCN | vanilla | 3.6539 | 66.7832 | 4.4214 | 91.2254 | 5.6993 | 134.4466 | 7.4967 | 198.6619 |
| | | w/ AGL | 3.5311(3.36%↓) | 66.2596(0.79%↓) | 4.3790(0.96%↓) | 95.7793(4.99%↑) | 5.9036(3.58%↑) | 138.7884(3.23%↑) | 7.7486(3.36%↑) | 198.0429(0.31%↓) |
| | WAVGCRN | vanilla | 4.9907 | 82.6108 | 5.3720 | 111.7549 | 6.9338 | 160.7631 | 16.76613 | 509.1225 |
| | | w/ AGL | 3.6849(26.16%↓) | 67.4041(18.41%↓) | 4.6048(14.28%↓) | 94.2841(15.63%↓) | 6.7905(2.07%↓) | 149.5729(6.96%↓) | 7.2698(56.64%↓) | 196.2354(61.47%↓) |
| PEMS08 | GWNet | vanilla | 15.8843 | 830.0247 | 18.3152 | 1196.2059 | 21.1694 | 1745.7639 | 23.8123 | 2278.5420 |
| | | w/ AGL | 15.5646(2.01%↓) | 826.3616(0.44%↓) | 17.9393(2.05%↓) | 1197.1906(0.08%↑) | 20.8184(1.66%↓) | 1765.5440(1.13%↑) | 23.0820(3.07%↓) | 2052.7918(9.91%↓) |
| | PMC-GCN | vanilla | 15.9641 | 840.1459 | 18.3765 | 1205.8416 | 22.0897 | 1808.6214 | 26.4791 | 2530.5159 |
| | | w/ AGL | 15.6584(1.91%↓) | 817.4102(2.71%↓) | 18.0830(1.60%↓) | 1191.1047(1.22%↓) | 21.6807(1.85%↓) | 1778.2541(1.68%↓) | 26.1885(1.10%↓) | 2460.0098(2.79%↓) |
| | STIDGCN | vanilla | 15.3111 | 790.1659 | 17.5923 | 1201.1804 | 19.8736 | 1641.5082 | 23.8807 | 2240.3438 |
| | | w/ AGL | 15.2417(0.45%↓) | 786.2132(0.50%↓) | 17.3605(1.32%↓) | 1200.6130(0.05%↓) | 19.8284(0.27%↓) | 1677.8730(2.22%↑) | 22.1619(7.20%↓) | 2168.7754(3.19%↓) |
| | STGCN | vanilla | 18.1283 | 951.7638 | 20.3185 | 1306.2333 | 22.9451 | 1931.8759 | 28.2738 | 2874.3938 |
| | | w/ AGL | 18.1212(0.03%↓) | 921.6303(3.17%↓) | 20.0083(1.53%↓) | 1304.6440(0.12%↓) | 23.1401(0.85%↑) | 1924.5240(0.38%↓) | 28.8598(2.07%↑) | 2779.5898(3.30%↓) |
| | WAVGCRN | vanilla | 22.6586 | 1314.9849 | 21.0412 | 1359.0916 | 28.9435 | 2339.8194 | 34.5698 | 3212.1919 |
| | | w/ AGL | 18.2564(19.43%↓) | 1002.9889(23.73%↓) | 20.3240(3.41%↓) | 1291.0087(5.01%↓) | 20.3240(3.41%↓) | 1897.6740(18.90%↓) | 27.6968(19.89%↓) | 2561.0410(20.27%↓) |

Table 9: Ablation study for core modules: AGL and ASL. Trend and Season, respectively, denote the trend feature processing and seasonal feature processing in ASL. **Bolded values** indicate the best results, underlined values indicate the second-best results.

| Dataset | | model | | 120 minutes | | 180 minutes | | 240 minutes | |
|---|---|---|---|---|---|---|---|---|---|
| | AGL | ASL | | MAE | MSE | MAE | MSE | MAE | MSE |
| | | Trend | Season | | | | | | |
| METR-LA | √ | √ | √ | **7.1034** | **191.6066** | **8.1956** | **224.3926** | **8.6342** | **249.3086** |
| | √ | √ | × | 7.7387 | 204.8815 | 8.8438 | 235.1586 | 9.4877 | 255.3363 |
| | √ | × | √ | 7.6340 | 194.6127 | 8.6864 | 230.1645 | 9.2585 | 249.8750 |
| | √ | × | × | 8.2673 | 213.8754 | 9.3672 | 253.5769 | 9.6361 | 278.6919 |
| | × | √ | √ | 8.0057 | 215.9630 | 9.1913 | 245.3591 | 9.9757 | 287.2614 |
| | × | √ | × | 8.3199 | 214.2154 | 9.4653 | 254.7740 | 10.3952 | 290.2376 |
| | × | × | √ | 8.0788 | 212.7841 | 9.0339 | 223.5677 | 10.1710 | 280.9923 |
| | × | × | × | 8.4021 | 223.9778 | 9.5560 | 257.2167 | 10.4587 | 290.7861 |

window of the simulated node, the node sequence is generated using a cosine function (Woo et al., 2022):

$$Node_1^i = \alpha_1 \cos(2\pi f_1(\boldsymbol{t}+i)) + \alpha_2 \cos(2\pi f_2(\boldsymbol{t}+i)) + \epsilon_1, \tag{13}$$

$$Node_2^i = \cos(\frac{2\pi}{900}i + s) * Node_1^i + \epsilon_2, \tag{14}$$

where $\alpha_1, \alpha_2 = 0.15$, $f_1 = \frac{1}{10}$, $f_2 = \frac{1}{13}$, and $s = \frac{\pi}{4}$. $\epsilon_1$ and $\epsilon_2$ are noise terms sampled from a Gaussian distribution with mean 0 and variance 0.005. $\boldsymbol{t} = [0, 1, \cdots, L+O] \in \mathbb{R}^{L+O}$ denotes the input time sequence. Following this setting, $Node_1$ and $Node_2$ exhibit a clearly defined seasonal relationship.

## A.8 ABLATION STUDY

In this section, we perform ablation studies to assess the contribution of DPGNet's core modules: Adaptive Graph Learner(AGL) and Adaptive Season Learner(ASL). Table 9 presents the results of the ablation study. In addition, we further assess the contribution of trend feature processing and seasonal feature processing in ASL. All core modules within DPGNet demonstrate their importance, as evidenced by a noticeable decline in prediction accuracy when any of these components is removed.

## A.9 SENSITIVITY ANALYSIS

In this section, we conduct a sensitivity analysis of the multi-scale parameter $m$. Table 10 presents the results of the sensitivity analysis for the multi-scale parameter $m$. As shown in the table, the multi-scale processing approach enhanced the prediction accuracy of DPGNet. Moreover, in the majority of cases, setting $m = 1$ yielded the best overall performance of the model.

## A.10 LIMITATIONS

In the following, we briefly discuss the limitations of DPGNet's core modules, AGL and ASL.

Table 10: Sensitivity analysis of the multi-scale parameter $m$ on METR-LA and Weather. **Bolded values** indicate the best results, underlined values indicate the second-best results.

| Dateset | Model | 120 minutes | | 180 minutes | | 240 minutes | |
|---|---|---|---|---|---|---|---|
| | | MAE | MSE | MAE | MSE | MAE | MSE |
| METR-LA | $m = 0$ | 8.0352 | 215.9630 | 8.7688 | 229.228 | 9.6037 | 259.0429 |
| | $m = 1$ | **7.1034** | **191.6066** | 8.1956 | 224.3926 | **8.6342** | **249.3086** |
| | $m = 2$ | 7.6167 | 194.5623 | **7.8676** | **223.1617** | 9.1886 | 257.1401 |
| | $m = 3$ | 7.2794 | 193.0593 | 8.5363 | 222.5458 | 9.9809 | 269.3656 |
| Weather | $m = 0$ | 0.1867 | 0.1119 | 0.2206 | 0.1443 | 0.2678 | 0.1824 |
| | $m = 1$ | 0.1653 | 0.1019 | **0.1969** | 0.1267 | **0.2136** | **0.1437** |
| | $m = 2$ | 0.1825 | 0.1120 | 0.2043 | 0.1347 | 0.2490 | 0.1590 |
| | $m = 3$ | **0.1624** | **0.1009** | 0.2101 | 0.1377 | 0.2400 | 0.1601 |
| | $m = 4$ | 0.1803 | 0.1124 | 0.2061 | 0.1400 | 0.2474 | 0.1708 |
| | $m = 5$ | 0.1672 | 0.1058 | 0.2031 | **0.1224** | 0.2640 | 0.1716 |

For AGL, while the patch processing approach significantly reduces the number of G-RNNs, the number still increases with the length of the input sequence. An excessive number of G-RNNs can lead to issues such as gradient explosion and high computational cost. Additionally, although AGL can handle weak connections between nodes, as demonstrated by the experiments in Section 3, it lacks the capability to extract useful information from these weak connections.

For ASL, its multi-scale framework improves the model's long-term prediction ability. However, ASL lacks an automated mechanism for selecting the multi-scale parameter $m$.

THE USE OF LARGE LANGUAGE MODELS

In this study, large language models (LLMs) were solely used for text refinement purposes. No other applications or functions of LLMs were utilized in the research.