# OpenReview forum: "DPGNet: Modeling Dynamic Graphs and Complex Temporal Patterns for Spatiotemporal Forecasting"
_ICLR.cc/2026/Conference — Submitted to ICLR 2026_

### Official Review · Reviewer_muoK · 2025-10-29

**Soundness:** 2
**Presentation:** 2
**Contribution:** 2
**Rating:** 4
**Confidence:** 4

**Summary:**

This work proposes the Dynamic Graph Prediction Network (DPGNet), which includes two key components: the Adaptive Graph Learner
(AGL) and the Adaptive Season Learner (ASL). AGL is a plug-and-play module that can effectively capture dynamic inter-node relationships while suppressing weak connections. ASL can model the complex temporal patterns of nodes and construct graph structures across different temporal patterns and time scales.

**Strengths:**

1.	The paper addresses an important problem in spatiotemporal prediction and highlights the issue of weak connections in learned graphs, which is relevant to the research community.
2.	The experimental evaluation covers multiple widely used benchmark datasets (e.g., METR-LA, PEMS08, PEMS-Bay), which allows for a fair comparison with existing methods.

**Weaknesses:**

1. There are inconsistencies in the experimental reporting. For example, in Section 4 the input/output lengths are described as O=3,6,12, but Table 1 uses O=3 while Table 3 reports “15min” and “60min.” Although these are equivalent, it is better to maintain consistent notation across the paper to avoid confusion. In addition, in Table 1 the reported STIDGCN result on PEMS08 for O=3 is 15.3111, whereas your proposed method achieves 15.2474. However, in Table 3, after adding AGL, STIDGCN achieves 15.2417, which is better than the best result reported in Table 1. This discrepancy is confusing and raises the question of whether the baseline results are fully optimized and whether the proposed improvement is robust. Please clarify the evaluation setting and ensure consistency in reporting.
2. Both the Adaptive Graph Learner (AGL) and the Adaptive Season Learner (ASL) lack sufficient novelty. The proposed designs appear to primarily reorganize and reframe existing methods rather than introduce fundamentally new mechanisms.
3. The experimental results show only limited improvements over existing baselines.
4. The explanation of Figure 2 is unclear. It is difficult for the reader to immediately understand what weak/strong connections mean, why the zero entries are separated, and what the key takeaway is. I recommend that the authors improve the figure caption, highlight the main conclusion, and consider presenting a simpler comparison (weak vs strong ratios) for clarity. Figure 2 is difficult to interpret. For instance, the bar corresponding to 7.47% appears only slightly lower than the bar corresponding to over 90%, which is counter-intuitive. The current visualization does not clearly convey the magnitude of difference between weak and strong connections. I recommend the authors revise the figure or choose a clearer representation.
5. The notation in the overview section is inconsistent with the symbols used in the corresponding figures. The use of boldface for mathematical symbols is not applied consistently (sometimes bold, sometimes not).

**Questions:**

1. There are inconsistencies in the experimental reporting. For example, in Section 4 the input/output lengths are described as O=3,6,12, but Table 1 uses O=3 while Table 3 reports “15min” and “60min.” Although these are equivalent, it is better to maintain consistent notation across the paper to avoid confusion. In addition, in Table 1 the reported STIDGCN result on PEMS08 for O=3 is 15.3111, whereas your proposed method achieves 15.2474. However, in Table 3, after adding AGL, STIDGCN achieves 15.2417, which is better than the best result reported in Table 1. This discrepancy is confusing and raises the question of whether the baseline results are fully optimized and whether the proposed improvement is robust. Please clarify the evaluation setting and ensure consistency in reporting.
2. The explanation of Figure 2 is unclear. It is difficult for the reader to immediately understand what weak/strong connections mean, why the zero entries are separated, and what the key takeaway is. I recommend that the authors improve the figure caption, highlight the main conclusion, and consider presenting a simpler comparison (weak vs strong ratios) for clarity. Figure 2 is difficult to interpret. For instance, the bar corresponding to 7.47% appears only slightly lower than the bar corresponding to over 90%, which is counter-intuitive. The current visualization does not clearly convey the magnitude of difference between weak and strong connections. I recommend the authors revise the figure or choose a clearer representation.

---

### Official Review · Reviewer_XUJY · 2025-10-31

**Soundness:** 3
**Presentation:** 3
**Contribution:** 1
**Rating:** 2
**Confidence:** 5

**Summary:**

This work proposes a novel spatio-temporal forecasting model to address limitations in capturing dynamic node relationships and complex temporal patterns.

**Strengths:**

- The framework is straightforward and easy to follow.
- The experiment is comprehensive.

**Weaknesses:**

- This work is with little novelty and fails to provide new perspectives. Constructing dynamic graphs or decomposing time series are both widely used strategies. And the proposed framework is merely a collection of existing commonly used models.
- The reported performance of baselines is inconsistent with existing literture. For instance, with horizon as 12, the MAE of STIDGCN  on PEMS08 in its origin paper is reported as $13.45$, less than $19.5711$ in this paper.

**Questions:**

See Weaknesses.

---

### Official Review · Reviewer_Dz25 · 2025-11-01

**Soundness:** 2
**Presentation:** 3
**Contribution:** 1
**Rating:** 0
**Confidence:** 4

**Summary:**

The paper proposes a new spatiotemporal forecasting model architecture by combining a dynamic graph learning module with a temporal module that learns at different time scales. The author then validates the model on several spatiotemporal prediction datasets. The AGL is also supplemented to baseline methods to show prediction improvements.

**Strengths:**

- Clarity: The paper provides clear and intuitive figures that show the model architecture and traffic data behaviour.
- Quality: The proposed model is benchmarked on several datasets in different domains, e.g. traffic, weather, and electricity. The model's graph learning module is also substituted into baseline methods to show performance improvement.

**Weaknesses:**

- Literature Review:
  - Decomposing temporal dynamics into different scales is also proposed in works such as ASTGCN by Guo, S., Lin, Y., Feng, N., Song, C., & Wan, H. (2019).
  - Section 2.2 is very sparse and deserves more discussion to place the paper in the context of existing work.
  - Several citations are pointing to the wrong original paper. For instance, vanishing gradient was more formally analyzed by
Pascanu, R., Mikolov, T., & Bengio, Y..(2013), and dilated convolutions were proposed by Yu, F., & Koltun, V. (2015).
- Experimental Setup:
  - The entire appendix is missing to provide more information on the experimental setup.
  - The baseline MAE and RMSE are reported to be lower in the original papers.

**Questions:**

- Why is MAPE not included in the metrics?
- What is the motivation behind using G-RNN? Since it operates on the adjacency matrix, how is it different from using GCNs interleaved with RNNs?
- Please provide appendix for detailed experimental setup.

---

### Official Review · Reviewer_r4TP · 2025-11-02

**Soundness:** 2
**Presentation:** 2
**Contribution:** 1
**Rating:** 2
**Confidence:** 5

**Summary:**

A complicated, composite method for spatiotemporal forecasting.

**Strengths:**

1. This work appears to be a combination of multiple previously proposed methods. Methodologically, I do not see anything new.
2. The choice of datasets is reasonable. It is good that the experiments do not use the ETT dataset, because this paper focuses on spatiotemporal forecasting, where it is more appropriate to use datasets with a large number of spatial nodes.

**Weaknesses:**

1. The motivation for introducing each component is not clearly explained. This is often the case in incremental work: the significance of each component is usually made explicit when it first appears in the original paper that proposed it. Here, since this method is formed by combining several prior methods, many steps are essentially skipped, which makes the overall architecture look complicated and not very elegant.
2. The ablation study is too simple. If this paper aims to meet the ICLR bar, it needs to demonstrate that every component works, I mean even down to details like the introducing of FC layer in the G-RNN. Otherwise, this paper will not provide meaningful insights to the community and will only waste the limited time of the authors, readers, and reviewers.

**Questions:**

same as weaknesses

---

### Meta-Review · Area_Chair_sRsW · 2026-01-02

**Summary:**

The reviewers expressed significant concerns about the submission, particularly regarding its lack of novelty and clarity in the spatiotemporal forecasting. Reviewer r4TP noted that the work appears to merely combine existing methods without substantial innovation, criticizing the unclear motivation for the model's components and the insufficient ablation study. Reviewer Dz25 highlighted deficiencies in the literature review, missing key metrics, and a lack of detail in the experimental setup. Reviewer XUJY reiterated concerns about the framework lacking originality, while Reviewer muoK pointed out inconsistencies in experimental results and questioned the novelty of the proposed components.

**Reviewer Concerns:**

The authors did not provide any rebuttals to address the reviewers' concerns.

**Reviewer Scores:**

Because the author did not provide any rebuttal, the reviewers are unable to engage in meaningful discussion with them.

---

### Decision · Program_Chairs · 2026-01-26

Reject